# Deep Learning Histology for Prediction of Lymph Node Metastases and Tumor Regression after Neoadjuvant FLOT Therapy of Gastroesophageal Adenocarcinoma

**DOI:** 10.3390/cancers16132445

**Published:** 2024-07-03

**Authors:** Jin-On Jung, Juan I. Pisula, Xenia Beyerlein, Leandra Lukomski, Karl Knipper, Aram P. Abu Hejleh, Hans F. Fuchs, Yuri Tolkach, Seung-Hun Chon, Henrik Nienhüser, Markus W. Büchler, Christiane J. Bruns, Alexander Quaas, Katarzyna Bozek, Felix Popp, Thomas Schmidt

**Affiliations:** 1Department of General, Visceral, Tumor and Transplantation Surgery, University Hospital of Cologne, Kerpener Straße 62, 50937 Cologne, Germany; 2Department of General, Visceral and Transplantation Surgery, University Hospital of Heidelberg, Im Neuenheimer Feld 420, 69120 Heidelberg, Germany; 3Data Science of Bioimages Lab, Center for Molecular Medicine Cologne (CMMC), Faculty of Medicine, University Hospital of Cologne, Robert-Koch-Straße 21, 50937 Cologne, Germany; 4Institute of Pathology, University Hospital of Cologne, 50937 Cologne, Germany

**Keywords:** artificial intelligence, deep learning, gastroesophageal cancer, chemotherapy response, FLOT therapy, prediction algorithm, neural network

## Abstract

**Simple Summary:**

The prediction of tumor response after neoadjuvant FLOT therapy is highly necessary. The use of deep learning on gastroesophageal biopsies enabled us to extract predictive information. This prediction model could be easily applied in clinical decision making. Patients could avoid unnecessary treatment or receive an intensified FLOT therapy.

**Abstract:**

Background: The aim of this study was to establish a deep learning prediction model for neoadjuvant FLOT chemotherapy response. The neural network utilized clinical data and visual information from whole-slide images (WSIs) of therapy-naïve gastroesophageal cancer biopsies. Methods: This study included 78 patients from the University Hospital of Cologne and 59 patients from the University Hospital of Heidelberg used as external validation. Results: After surgical resection, 33 patients from Cologne (42.3%) were ypN0 and 45 patients (57.7%) were ypN+, while 23 patients from Heidelberg (39.0%) were ypN0 and 36 patients (61.0%) were ypN+ (*p* = 0.695). The neural network had an accuracy of 92.1% to predict lymph node metastasis and the area under the curve (AUC) was 0.726. A total of 43 patients from Cologne (55.1%) had less than 50% residual vital tumor (RVT) compared to 34 patients from Heidelberg (57.6%, *p* = 0.955). The model was able to predict tumor regression with an error of ±14.1% and an AUC of 0.648. Conclusions: This study demonstrates that visual features extracted by deep learning from therapy-naïve biopsies of gastroesophageal adenocarcinomas correlate with positive lymph nodes and tumor regression. The results will be confirmed in prospective studies to achieve early allocation of patients to the most promising treatment.

## 1. Introduction

In the multimodal treatment of gastroesophageal adenocarcinoma, neoadjuvant therapy is currently the standard of practice prior to oncological resection. The FLOT4 study by Al-Batran et al. from 2019 demonstrated the superiority of the FLOT regimen over other chemotherapy protocols for both gastric carcinoma and adenocarcinoma of the gastroesophageal junction [1]. However, not all patients benefit to the same extent from FLOT therapy and individual response rates can vary highly. In a study by Donlon et al. including 175 patients with cT2 to cT4a and any cN cases of gastroesophageal adenocarcinoma, 54% of patients had remaining positive lymph nodes (ypN+) after neoadjuvant FLOT therapy and successive oncological resection in the final histological workup [2]. Furthermore, there were variations in residual vital tumor (RVT) as a fraction of the whole tumor site. Schmidt et al. reported a partial or complete response in 49.8% of patients receiving neoadjuvant chemotherapy [3]. Therefore, it is essential to know if a foreseen neoadjuvant therapy according to the FLOT protocol would result in cases of ypN0 or ypN+ in the final histopathological workup. So far, the prediction of FLOT response has not been covered intensively in the current literature, which has mainly focused on adjuvant chemotherapy response [4] or worked with hematological [5] and nutritional [6] biomarkers with poor prognostic power.

### 1.1. Lymph Node Metastases as Outcome Parameter

Established prognostic criteria for oncological outcome in gastroesophageal carcinoma include TNM stage, tumor histology, the number of affected lymph nodes, R-status, postoperative complications, and tumor regression [7]. With the exception of postoperative complications, all prognostic criteria are directly or indirectly related to tumor response after neoadjuvant therapy. Previous machine learning studies have also shown that lymph node status (pN) is one of the most important predictors for long-term overall survival in upper gastrointestinal cancer [8] surpassing tumor regression after CROSS therapy [9]. Therefore, if the potentially achievable regression of lymph node metastases could be predicted before administering neoadjuvant FLOT therapy, this could ultimately influence and improve long-term prognosis.

### 1.2. Deep Learning in Histology

The prediction of tumor characteristics based on digitized histological slides has been addressed extensively in recent years through the development of deep learning neural networks. Sang et al. demonstrated in 2021 that deep learning based on digitized whole-slide images enables the classification of lung cancer [10]. Furthermore, standard hematoxylin and eosin (H&E) staining of histological samples has been introduced as a promising biomarker using neural networks. Deep learning techniques have the potential to detect molecular tumor characteristics solely based on H&E staining. Thus, Kather et al. were able to predict microsatellite instability in gastrointestinal tumors directly from H&E staining [11]. Mobadersany et al. predicted overall survival in patients with gliomas based on histology and additional genomic analyses [12].

### 1.3. Aim of this Study

The aim of this study was to establish a deep learning prediction model using clinical information and visual data from digitized whole-slide images of biopsies used for initial diagnosis of gastroesophageal adenocarcinoma in therapy-naïve patients. All relevant data that were used for this study were thus present before neoadjuvant therapy was performed. The prediction model was developed to predict lymph node status (positive or negative) and therapeutic response to neoadjuvant FLOT therapy.

## 2. Materials and Methods

### 2.1. Patient Cohorts and Inclusion Criteria

Eligible subjects of this study were patients with esophageal or gastroesophageal junction adenocarcinoma who underwent abdominothoracic Ivor-Lewis esophagectomy with gastric tube reconstruction between January 2013 and May 2021 at the University Hospital of Cologne, Department of General Surgery (see Figure 1). To be considered for this retrospective analysis, only cases with full administration of four cycles of preoperative FLOT therapy and a complete histopathological workup after resection were included. Tumors that were staged as uT1 early cancer or uT4b were excluded from analysis. Furthermore, the developed algorithm was validated using an external validation cohort from the University Hospital of Heidelberg, Department of General Surgery, that met the same inclusion criteria.

To adjust for differences between the training and validation cohort, only patients with full documentation of clinical information were further selected for this study. A subset of 11 common clinical parameters resulted out of the 103 preoperative variables from the University Hospital Cologne and 79 preoperative variables from the University Hospital Heidelberg that were being systematically assessed. These clinical variables were namely sex, age, physical status according to the American Society of Anesthesiologists (ASA classification), body mass index (BMI), uT/cT status, uN/cN status, and histological grading. Moreover, relevant information from the past medical history (PMH) of the patients was analyzed. Cardiovascular PMH included arterial hypertension, myocardial infarction, or prior coronary angiography, pulmonary PMH included conditions such as bronchial asthma or chronic obstructive pulmonary disease (COPD), and metabolic PMH included diabetes mellitus, hypercholesteremia, or any other metabolic disorder. Any severe PMH was defined as one or a combination of the above-mentioned pre-existing conditions that compromised operability, leading to an increased ASA classification or perioperative risk. Eventually, it was necessary to select only pretherapeutic parameters before neoadjuvant therapy because preoperative parameters after administered FLOT therapy were not eligible for inclusion. Last but not least, the information had to be available at both centers in Cologne and Heidelberg.

The retrospective trial protocol was approved by the local Ethics Committee of the University of Cologne under vote number 23-1217. The analysis of the Heidelberg cohort was approved by the Ethical Review Committee of the University of Heidelberg under vote number S-635/2013. Data management was in accordance with the Declaration of Helsinki, Good Clinical Practices as well as local legal requirements.

### 2.2. Acquisition of Biopsies

Since the primary diagnoses were not always made at the University Hospital of Cologne where oncological resection was eventually performed, additional primary biopsies from external reference centers thus had to be requested. It was partly necessary to re-cut the requested paraffine blocks and to re-stain the biopsies with hematoxylin and eosin (H&E). These steps were performed in collaboration with the Institute of Pathology at the University Hospital of Cologne.

### 2.3. Scanning and Digitalization

The H&E-stained biopsies taken at the time of initial diagnosis were obtained from the Pathological Institute of the University of Cologne and external pathological reference centers to enlarge the dataset. The biopsies were digitized using the high-resolution NanoZoomer S360 digital slide scanner (model number C13220) from Hamamatsu Photonics (Shizuoka, Japan). The WSIs were scanned under 40× objective magnification and typical settings including a resolution of 0.230 µm/pixel and were saved in .ndpi format. The Heidelberg slides were scanned using Aperio CS2 by Leica Biosystems (Wetzlar, Germany) under 40× magnification and saved in .svs format.

### 2.4. Region of Interest and Preprocessing (Tessellation)

All slides were manually annotated by defining the region of interest (ROI) encircling the tumor area on the WSIs with biopsy tissue. The annotations were performed using QuPath 0.5.1 [13] and additional scripts for automatization. The Libvips library was used to process the images outside of QuPath. The automated preprocessing of regions before training included Macenko stain normalization [14], tiling the regions into a tile size of 299 pixels/100 µm, a gray-space fraction of 0.6 and a threshold of 0.05, a white-space fraction of 1.0 and a threshold of 230, and additional Otsu thresholding [15] for background exclusion. Figure 2a shows an example biopsy with the marked region of interest (ROI) and the resulting tessellation into 617 training tiles, whereas Figure 2b demonstrates the distribution of the amount of extracted tiles for the whole cohort, resulting in the total number of 342,545 tiles.

### 2.5. Computational Resources and Implementation

The computational resources of the high-performance computing cluster (CHEOPS) of the University of Cologne in collaboration with the Center for Molecular Medicine Cologne (CMMC) were used to train the neural networks. The computing nodes are equipped with 4 NVIDIA V100 Volta graphics processing units each. Data management as well as handling of the clinical datasets were implemented using the open-source library Python in version 3.9 [16]. The neural networks were trained on the platform Slideflow 2.3 developed by Dolezal et al. [17]. Slideflow implements both backends Tensorflow [18] and PyTorch [19] for handling WSI data and for working with various architectures and pretrained models.

### 2.6. Model Architecture and Hyperparameters

The slide-based neural network that was used in this study was designed with Xception by Google Inc. (Mountain View, CA, USA) as the basic model architecture. Xception is a convolutional neural network (CNN) that was shown to outperform its predecessor Inception (version 3) on the ImageNet dataset [20]. Xception was already used in various studies for the automated differentiation between benign and malign lesions of rectal cancer [21], gastric ulcers [22], hepatocellular nodular lesions [23], pulmonary nodules [24], and breast cancer [25,26].

The selection of hyperparameters was optimized by sweep searching numerous sets of hyperparameters integrated in Slideflow by Dolezal et al. [17]. Due to the binary outcome parameter of positive or negative lymph nodes, the loss function was targeted toward sparse categorical cross-entropy. Data augmentation as another potentially alterable hyperparameter was performed by randomly flipping the tiles along the *x*- and *y*-axis, and using random rotation and different normalization techniques. Eventually, the ideal hyperparameters for the lymph node model were determined in terms of a batch size of 48, learning rate of 5 × 10^−5^, number of epochs 3, and dropout rate of 0.1. The hyperparameters of the final neural networks are shown in Appendix A and the architecture with the number of layers and trainable parameters is demonstrated in Appendix A. As shown in Appendix A, the clinical parameters are called slide features and are added to the neural network after the post-convolution layer as a further input layer.

In supplementary analyses, the models were trained to predict tumor regression according to Becker grading [27]. For this purpose, the loss function was changed to sparse categorical cross-entropy to binarily classify different Becker groups but was directed toward the root mean squared error (RMSE) to train the network on the percentage of residual vital tumor (RVT) that is known for the Cologne patient collective. Since this information was not available for the Heidelberg group, the predicted percentage of RVT was translated into Becker groups and then compared binarily to patients with either more or less than 50% residual vital tumor. Furthermore, Slideflow offers the option to train multiple-instance learning (MIL) models with separate feature extraction and several architectures such as attention-based MIL [28], clustering-constrained attention MIL/CLAM [29], and transformer-based MIL models [30]. During all reported analyses, multiple testing configurations with consistent switching between different training and testing sets were tried.

## 3. Results

### 3.1. Data Overview

This study included 137 patients with a total number of 227 whole-slide images (WSIs) and a total file size of 77.3 gigabytes. A total of 78 patients belonged to the collective from the University Hospital of Cologne used as the training cohort, whereas 59 patients belonged to the collective from the University Hospital of Heidelberg used as the external validation cohort.

Table 1 summarizes the clinical input variables used for this study as well as the outcome parameters that were considered for different prediction models. There were significant differences regarding ASA classification indicating that the Cologne patients had lower grades than patients from Heidelberg (*p* < 0.001). In the Heidelberg cohort, there were significantly lower levels of histological grading (*p* = 0.028). Both cohorts did not differ significantly regarding sex, age, BMI, uT/cT status, cN status, and past medical history.

On average, 36.8 lymph nodes were resected in the Cologne cohort compared to 29.5 resected lymph nodes in the Heidelberg cohort (*p* < 0.001). Other than this potential outcome parameter to be predicted, there were no statistically significant differences between the two groups. In particular, the final outcome parameter ypN0 vs. ypN+ used for the reported neural network was similarly distributed between the two groups, with 33 cases (42.3%) of ypN0 and 45 cases (57.7%) of ypN+ in Cologne compared to 23 cases (39.0%) of ypN0 and 36 cases (61.0%) of ypN+ in Heidelberg (*p* = 0.695). Becker grades were not significantly different between both cohorts. In Cologne, 43 patients (55.1%) were graded as Becker 1a, 1b, or 2, whereas 35 patients (44.9%) were Becker grade 3. In Heidelberg, 34 patients (57.6%) were graded as Becker 1a, 1b, or 2, whereas 25 patients (42.4%) were Becker grade 3 (*p* = 0.955).

### 3.2. Performance of the Neural Networks

The neural network introduced in Section 2.6 (see above) was trained to separately classify tiles, slides, and patients only from the Cologne group into the binary outcome parameter positive or negative lymph nodes (ypN0 vs. ypN+) extracted from the final histopathological workup. At the beginning of training, the accuracy was calculated as 49.7% and reached 92.1% after 17,793 batches and 3 epochs (see Figure 3a). Likewise, the loss function started at 1.071 and reached 0.201 at the end of training. Residual vital tumor (RVT) was predicted using the mean squared error (MSE) as the loss function for training. The MSE started with values above 2200 and decreased after 29,655 batches and 5 epochs to 197.2, resulting in an error of ±14.1% for predicting the true percentage of RVT (see Figure 3b). For both models, the ideal batch size was determined as 48 after hyperparameter optimization and each epoch contained 5931 batches.

The successive external validation of the trained lymph node model on the Heidelberg dataset was able to achieve a patient-level area under the curve (AUC) of 0.698 when trained without the additional clinical dataset. For the same model with identical hyperparameters but trained with additional clinical information, the AUC could be improved to 0.726 (see Figure 4a).

Since the actual RVT fractions of the Heidelberg collective were not available (see Table 1), it was necessary to use the Becker grade as the validation metric. Becker grades 1a, 1b, and 2 are summarized as patients with an RVT of <50% and the remaining Becker grade 3 represented all patients with an RVT of >50% that was translated into a binary outcome parameter. Finally, external validation showed an AUC of 0.604 that could also be improved to an AUC of 0.648 when trained with clinical data (see Figure 4b).

### 3.3. Heatmap Visualization of Tile Importance

To interpret the neural networks, heatmaps were generated and analyzed for all biopsies that were used for prediction. Figure 5 demonstrates one primary biopsy in its native H&E staining and then again as a heatmap with all tiles colorized with red for high probability and blue for low probability according to the lymph node prediction model. After reviewing all biopsy heatmaps, the signal seems to come from the periphery of the specimen or, in other words, from the superficial tumorous parts of the tissue.

### 3.4. Weakly Supervised Training, Additional Clinical Endpoints, and Different Combinations of Training and Testing Groups

Further models were trained with various outcome parameters besides the already presented lymph nodes and percentage of RVT. The classical definition of major and minor regression was also used as a potential further outcome parameter with binary and ordinal outcome prediction. In contrast to the differentiation between an RVT of less or more than 50%, the typical major response is defined as Becker grades 1a or 1b (less than 10% residual vital tumor) and minor response is defined as Becker grades 2 or 3 (more than 10% residual vital tumor). Different loss functions for the models were also defined to explore potentially better performances. Moreover, multiple-instance learning (MIL) models were also implemented with various feature extractions and models, as described in Section 2.6. Finally, different configurations were assessed by combining various cohorts and patient groups. However, all results were not better than the presented slide-based models from Section 3.2.

## 4. Discussion

This interdisciplinary study unites patient data and methods from oncological surgery, pathology, and data science. The results suggest a correlation between visual information extracted by deep learning analyses of therapy-naïve biopsies of gastroesophageal adenocarcinomas and outcome parameters such as positive lymph nodes and tumor regression.

The model trained for the prediction of tumor regression (or residual vital tumor) demonstrated a mean squared error of 197.2% that can be translated by extracting the root into an eventual error of ±14.1%. However, this model’s performance was weaker in the external validation than in the prediction of ypN0 and ypN+ status. The area under the curve (AUC) of the lymph node model was 0.726 and the AUC of the tumor regression model was 0.648.

There is no comparable study so far that deals with the question of response prediction for gastroesophageal adenocarcinoma with a methodology such as the one presented in this study. Bremm et al. have tried to correlate tumor volume from computer tomography to distinguish FLOT and CROSS responders from non-responders [31]. Since there was no strong correlation, the authors state that other biological markers of prediction are urgently needed. Yoon et al. made use of patient-derived tumor organoids to mimic the original tumor and identify cases resistant to FLOT and FOLFOX therapy [32]. Although a promising approach, it may not be time- and cost-effective to gather this kind of predictive information.

### 4.1. Strengths and Limitations

This study features clinical parameters added to the neural networks and demonstrates that this inclusion outperforms the models trained without clinical parameters. In the current literature, there are only a few publications that report an integrative model of such kind. Another outstanding feature of this study is the rare possibility to externally validate the results from the University Hospital of Cologne with a patient collective from another high-volume medical center at the University Hospital of Heidelberg with identical input variables. The eventual AUC values may not seem convincing but are still of high value considering the fact that this kind of external validation occurs relatively rarely in the medical literature. A multicentric external validation cohort from various geographical regions would surely contribute to the generalizability of the model. However, it is still notable that in this case, validation was performed on a totally independent and different patient group.

One limitation of this study is the relatively small sample size of the training and testing cohort. On the other hand, data curation focused on complete cases and complete information, which may have contributed to the eventual case numbers. Most importantly, the outcome parameters ypN0/ypN+ status and minor/major response were balanced in both groups. Surely, there are more clinical variables such as local tumor length or circumference that could be used as further input parameters in an even bigger model. However, this was not possible because these kinds of data were not fully available in the presented cohorts. One more limitation comes from the clinical conclusions that can be drawn from these results at this point. Regardless of performance metrics, it may be too early to interpret this study in a way that a predicted low FLOT response would restrain clinicians from administering FLOT therapy in the future. It is possible that even though patients had a ypN+ status or more than 50% residual vital tumor in the final pathology, they would have resulted in an even worse tumor if not treated by FLOT therapy.

### 4.2. Perspective

These first results should be confirmed in prospective studies and could enable an early allocation of patients to the best possible neoadjuvant therapy, ultimately improving the oncological outcome in the future. In fact, a prospective setting is currently allocating a future patient collective at the University Hospital of Cologne to further examine a model trained with data from both centers, Heidelberg and Cologne. Until now, the validation dataset from Heidelberg was excluded from training to make sure that external validation was not biased. The new model resulting from both datasets could have a better understanding of the entirety of biopsies since more types will be included.

Currently, there is a general trend toward personalized therapy in the oncological treatment of gastroesophageal cancer. For instance, recent studies suggest that the administration of immune checkpoint inhibitor nivolumab in advanced cases of gastroesophageal adenocarcinoma should be individualized based on PD-L1 expression [33]. In the future, it can be assumed that neoadjuvant treatment will also have to adapt to the individual and patient-specific profile. Eventually, it is necessary to discuss the two therapeutic regimens of CROSS and FLOT that are currently competing in neoadjuvant gastroesophageal cancer therapy. A direct comparison of these two competing treatment regimens is still needed and the results of the ongoing randomized controlled ESOPEC trial are urgently awaited [34].

## 5. Conclusions

This study demonstrates how visual features can be extracted by deep learning histology from therapy-naïve biopsies of gastroesophageal adenocarcinoma. In combination with clinical parameters, the trained neural network was able to correlate the histological information with positive lymph nodes and tumor regression from the final histopathology. The established prediction model for tumor response after neoadjuvant FLOT therapy could thus individualize and improve oncological treatment in the future. However, the results will have to be confirmed in prospective studies to achieve early allocation of patients to the most promising treatment.

## Figures and Tables

**Figure 1 cancers-16-02445-f001:**
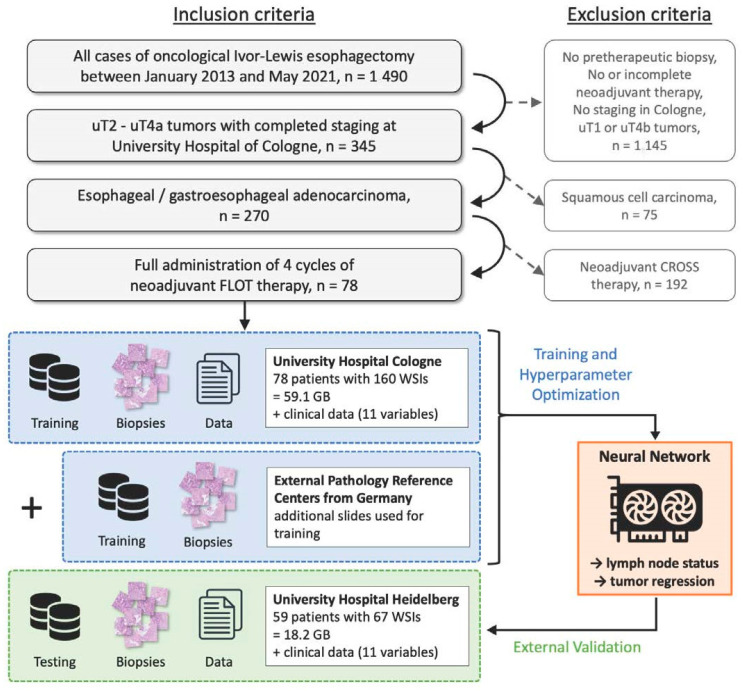
The study design as a workflow to demonstrate the different sources of the primary biopsies. Abbreviations: WSI = whole-slide image, and GB = gigabyte.

**Figure 2 cancers-16-02445-f002:**
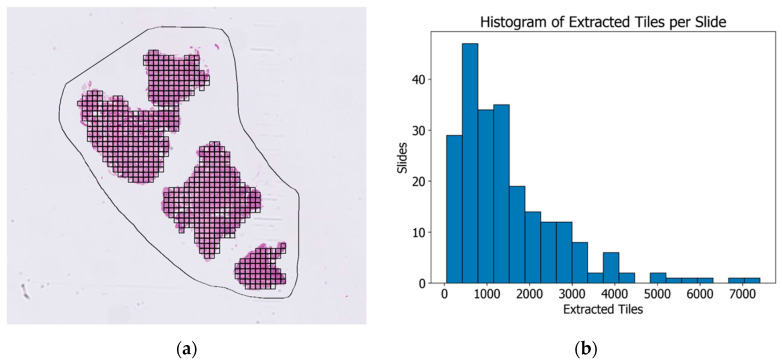
An example of a primary biopsy sample with the manually annotated region of interest (ROI, marked with a black line) and successive automated tessellation to 617 tiles (patches) (**a**). The total number of training tiles for this project was 342,545; the distribution of extracted tiles per WSI is shown in (**b**).

**Figure 3 cancers-16-02445-f003:**
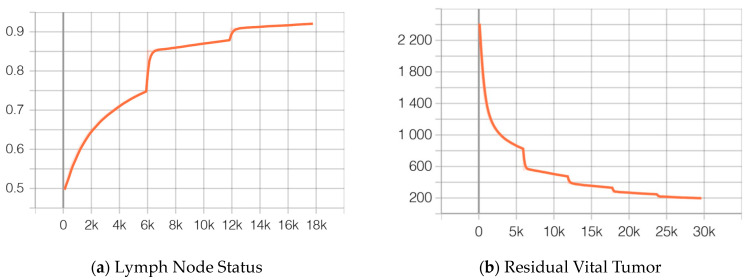
Increments in accuracy (**a**) up to 92.1% for the classification of positive vs. negative lymph nodes during the training phase. The training took a total of 17,793 batches and 3 epochs. Likewise, the model trained to predict the degree of residual vital tumor (**b**) declined toward a mean squared error of 197.2 after 29,655 batches and 5 epochs.

**Figure 4 cancers-16-02445-f004:**
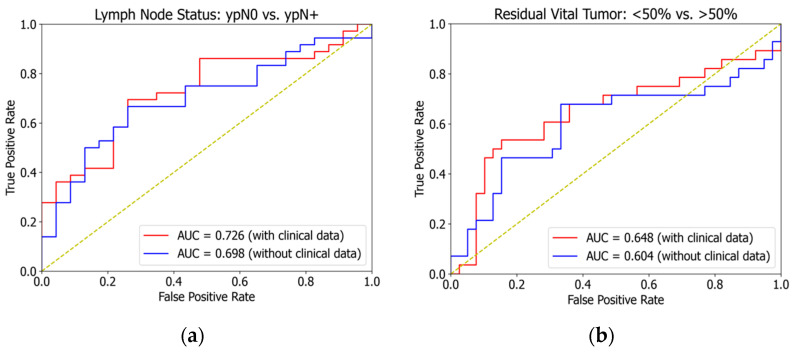
External validation with an area under the curve (AUC) of 0.726 for the binary classification of negative lymph nodes ypN0 vs. any positive lymph nodes ypN+ in the final histology (red curve (**a**)) compared to 0.698 when trained without clinical data (blue curve (**a**)). The AUCs for binarily classifying major responders with a residual vital tumor of <50% vs. minor responders with a residual vital tumor >50% were 0.648 (red curve (**b**)) and 0.604 (blue curve (**b**)) when trained without clinical data. The diagonal yellow line demonstrates a random prediction with an AUC of 0.5.

**Figure 5 cancers-16-02445-f005:**
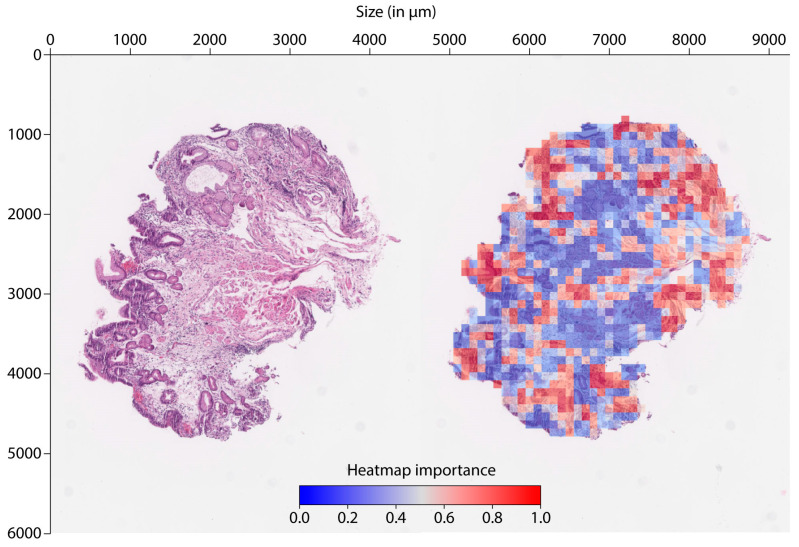
Normal H&E-stained, therapy-naïve histology slide and its correlating heatmap depicting the tiles with higher values close to 1.0 (marked in red) that were predictive for positive lymph nodes.

**Table 1 cancers-16-02445-t001:** An overview and comparison of the 11 clinical parameters that were used as additional input together with the biopsy slides for training the neural network. Presented are also the potential outcome parameters in the lower part of the table, with the final binary outcome parameter ypN0 vs. ypN+ and residual vital tumor/Becker grade marked in green. Abbreviations: ASA = American Society of Anesthesiologists, BMI = body mass index, PMH = past medical history, [X-Y] = 95% confidence interval, N/A = not available, ^†^ = chi-square test, and ^#^ = *t*-test.

Clinical Input Variables	Cologne Patients (*n* = 78)	Heidelberg Patients (*n* = 59)	*p*-Value
Sex—male vs. female	69 (88.5%), 9 (11.5%)	51 (86.4%), 8 (13.6%)	0.722 ^†^
Age (in years)	mean: 61.3 [59.4–63.1]	mean: 60.9 [58.0–63.9]	0.826 ^#^
ASA classification—1, 2 vs. 3	20 (25.6%), 49 (62.8%), 9 (11.5%)	1 (1.7%), 33 (55.9%), 25 (42.4%)	**<0.001 ** ^†^
BMI (in kg/m^2^)	mean: 27.7 [26.6–28.7]	mean: 26.1 [24.7–27.5]	0.071 ^#^
uT/cT status—T2, T3 vs. T4	9 (11.5%), 64 (82.1%), 5 (6.4%)	6 (10.2%), 50 (84.7%), 3 (5.1%)	0.910 ^†^
cN status—cN0 vs. cN+	8 (10.3%), 70 (89.7%)	5 (8.5%), 54 (91.5%)	0.725 ^†^
Grading—G1, G2 vs. G3	0 (0.0%), 31 (39.7%), 47 (60.3%)	5 (8.5%), 24 (40.7%), 30 (50.8%)	**0.028** ^†^
Any severe PMH—yes vs. no	15 (19.2%), 63 (80.8%)	20 (33.9%), 39 (66.1%)	0.051 ^†^
Cardiovascular PMH—yes vs. no	46 (59.0%), 32 (41.0%)	28 (47.5%), 31 (52.5%)	0.180 ^†^
Pulmonary PMH—yes vs. no	10 (12.8%), 68 (87.2%)	9 (15.3%), 50 (84.7%)	0.683 ^†^
Metabolic PMH—yes vs. no	14 (17.9%), 64 (82.1%)	12 (20.3%), 47 (79.7%)	0.724 ^†^
**Outcome variables**			
ypT status—ypT0/1/2 vs. ypT3/4	33 (42.3%), 45 (57.7%)	19 (32.2%), 40 (67.8%)	0.733 ^†^
ypN status—ypN0 vs. ypN+	33 (42.3%), 45 (57.7%)	23 (39.0%), 36 (61.0%)	0.695 ^†^
Number of positive lymph nodes	mean: 4.1 [2.5–5.8]	mean: 3.7 [1.9–5.5]	0.732 ^#^
Number of resected lymph nodes	mean: 36.8 [33.8–39.8]	mean: 29.5 [26.8–32.1]	**<0.001** ^#^
Ratio pos./all lymph nodes (in %)	mean: 10.1 [6.3–14.0]	mean: 11.8 [6.9–16.7]	0.588 ^#^
Becker grade—1a/1b, 2 vs. 3	18 (23.1%), 25 (32.1%), 35 (44.9%)	16 (27.1%), 18 (30.5%), 25 (42.4%)	0.955 ^†^
Residual vital tumor (in %)	mean: 43.9 [36.4–51.3]	N/A	N/A

## Data Availability

The data that support the findings of this study are available on request from the corresponding author, J.-O.J. The data are not publicly available due to privacy restrictions.

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
