# Peer review of "Deep Learning Histology for Prediction of Lymph Node Metastases and Tumor Regression after Neoadjuvant FLOT Therapy of Gastroesophageal Adenocarcinoma"

_cancers, 2024, doi:10.3390/cancers16132445_

Round 1

Reviewer 1 Report

Comments and Suggestions for Authors
Previous study establish a deep learning prediction model using WSI of biopsy sample. Using biopsy sample images to predict positive lymph nodes and tumor regression is very meaningful for clinical decision making. Despite the advantages of the study, I have several questions.

1. Sample size is a little small and with unbalanced characteristics in training and test datasets.

2. A concrete flowchart of inclusion and exclusion of patients is need.

3. How do you choose the 11 clinical parameters? Based on your experience? Or other reasons?

4. Residual vital tumor was N/A in test dataset which may cause potential bias.

5. The AUC of the model for predicting lymph nodes status and residual vital tumor is relatively low and not convincing. I wonder what will happened if you only train the model with clinical parameters?

Comments on the Quality of English Language

The quality of English is well.

Reviewer 2 Report

Comments and Suggestions for Authors

In this study, a deep learning model was developed to predict lymph node metastases and tumor regression in gastroesophageal adenocarcinoma following neoadjuvant FLOT treatment. In the research, a neural network model was created from biopsy samples obtained from a total of 137 patients, 78 from Cologne University Hospital and 59 from Heidelberg University Hospital. The model provides 92.1% accuracy in predicting lymph node metastases and shows that it predicts tumor regression with an error rate of +/- 14.1%. The deep learning model predicts lymph node metastases with 92.1% accuracy. The usability of the model in optimizing treatment strategies provides practical benefit in clinical decision-making processes. In this sense, I believe that the study will make a significant contribution to the literature. However, I recommend that you kindly consider the following gentle suggestions.

1-The study was conducted with a limited number of patient data and would be more effective if it could be generalized to larger populations.

2-External validation from the University of Heidelberg could not fully establish whether the model performs the same in every hospital, this should be clearly stated

3-The validity of the model has not been fully tested with data sets from different geographical regions and demographic groups, these tests should be added.

4-The complexity of the neural network model requires technical knowledge and resources for its use in clinical settings. The model is trained using specific clinical parameters and other potential clinical data must be included in the model.

5- The conclusion section of the study is written very briefly and should be expanded in detail.

6- Although a discussion section has been written, I recommend making comparisons with studies in the literature.

7- The horizontal thorn axis unit in Figure 5 should be written (2000?, 3000?)

8- The terms vital tumor and lymph nodes in Figure 4 should be explained in detail and the tolerance value should also be explained.

Round 2

Reviewer 1 Report

Comments and Suggestions for Authors  I have reviewed the revised manuscript with the response letter, and  after the revision, the present paper has been sufficiently improved to publication.